# Automated differentiation of wide QRS complex tachycardia using QRS complex polarity
Adam M. May[1] ✉, Bhavesh B. Katbamna[2], Preet A. Shaikh[1], Sarah LoCoco[1], Elena Deych[3], Ruiwen Zhou[3], Lei Liu[3], Krasimira M. Mikhova[1], Rugheed Ghadban[1], Phillip S. Cuculich[1], Daniel H. Cooper[1], Thomas M. Maddox[1], Peter A. Noseworthy[4] & Anthony Kashou[4]

## Abstract

**Background** Wide QRS complex tachycardia (WCT) differentiation into ventricular tachycardia (VT) and supraventricular wide complex tachycardia (SWCT) remains challenging despite numerous 12-lead electrocardiogram (ECG) criteria and algorithms. Automated solutions leveraging computerized ECG interpretation (CEI) measurements and engineered features offer practical ways to improve diagnostic accuracy. We propose automated algorithms based on (i) WCT QRS polarity direction (WCT Polarity Code [WCT-PC]) and (ii) QRS polarity shifts between WCT and baseline ECGs (QRS Polarity Shift [QRS-PS]).
**Methods** In a three-part study, we derive and validate machine learning (ML) models—logistic regression (LR), artificial neural network (ANN), Random Forests (RF), support vector machine (SVM), and ensemble learning (EL)—using engineered (WCT-PC and QRS-PS) and previously established WCT differentiation features. Part 1 uses WCT ECG measurements alone, Part 2 pairs WCT and baseline ECG features, and Part 3 combines all features used in Parts 1 and 2
**Results** Among 235 WCT patients (158 SWCT, 77 VT), 103 had gold standard diagnoses. Part 1 models achieved AUCs of 0.86–0.88 using WCT ECG features alone. Part 2 improved accuracy with paired ECGs (AUCs 0.90–0.93). Part 3 showed variable results (AUC 0.72–0.93), with RF and SVM performing best.
**Conclusions** Incorporating engineered parameters related to QRS polarity direction and shifts can yield effective WCT differentiation, presenting a promising approach for automated CEI algorithms.

## Plain language summary

Wide QRS complex tachycardias (WCTs) are abnormal, rapid heart rhythms that can be dangerous. Differentiating between the two main types, which are ventricular tachycardia (VT) and supraventricular wide complex tachycardia (SWCT), is critical for treatment decisions but remains challenging. An electrocardiogram (ECG) measures the electrical activity of the heart. We used automated ECG measurements to develop computational methods that enhance the accuracy of ECG interpretation. The computational methods, particularly those that analyzed paired ECG recordings, were able to differentiate WCTs with high accuracy. This method could help doctors diagnose heart conditions more reliably, resulting in faster and more precise treatments for patients with abnormal heart rhythms.

Wide QRS complex tachycardia (WCT) is defined as a rapid rhythm with a ventricular rate greater than 100 beats per minute (bpm) and QRS duration greater than 120 ms. The attributable causes of WCT include ventricular tachycardia (VT), supraventricular wide complex tachycardia (SWCT) due to pre-existing or functional aberrancy, SWCT arising from impulse propagation over atrioventricular accessory pathways (i.e., pre-excitation), tachycardias occurring with coexisting toxic-metabolic QRS duration widening (e.g., hyperkalemia), and rapid ventricular pacing. It is critical that patient-facing clinicians accurately and promptly determine whether the

WCT is due to VT or SWCT as there are important implications pertaining to immediate patient care decisions, subsequent clinical workup, and long-term management strategies.

The need for diagnostic tools to help clinicians accurately discriminate WCTs has been recognized for several decades[1,2]. Numerous manual 12-lead electrocardiogram (ECG) interpretation algorithms and criteria have been developed to help clinicians distinguish VT from SWCT[3–14]. While manual algorithms have generally demonstrated favorable diagnostic performance when applied by heart rhythm experts within highly regulated

[1]Department of Medicine, Division of Cardiovascular Diseases, Washington University School of Medicine in St. Louis, St. Louis, MO, USA. [2]Division of Cardiovascular Diseases, Loyola University Chicago, Stritch School of Medicine, Maywood, IL, USA. [3]Division of Biostatistics, Washington University School of Medicine in St. Louis, St. Louis, MO, USA. [4]Department of Cardiovascular Medicine, Mayo Clinic, Rochester, MN, USA. ✉e-mail: may.adam@wustl.edu

research settings, manual methods retain several important limitations when they are considered for broader clinical use. First, the diagnostic performance of manual methods is inextricably dependent on ECG interpreter competence and experience, which can be robust or lacking[15-18]. Second, the unpracticed application of manual algorithms can be a problematic and time-consuming task for non-experts, one that is especially challenging for clinicians responsible for managing high-acuity and/or clinically deteriorating patients.

In order to counter the practical diagnostic limitations of manual algorithms, as well as supplement their known strengths, we previously developed and validated automated WCT differentiation algorithms using computerized ECG interpretation (CEI) software[19-23]. The core aspect of automated approaches lies in the utilization of readily available computerized ECG measurements (such as WCT QRS duration) and innovative customized features (like horizontal and frontal percent amplitude change [PAC]) for the differentiation of VT and SWCT through machine learning (ML) modeling techniques[24].

In this context, herein we introduce WCT differentiation algorithms that harness new innovative features and ML modeling techniques. Specifically, we introduce ML models that utilize features characterizing the direction of the WCT QRS complex (referred to as WCT Polarity Code [WCT-PC]) and directional shifts in the QRS complex between the WCT and the baseline ECG (referred to as QRS Polarity Shift [QRS-PS]) among leads comprising the standard 12-lead ECG.

## Methods

### Study design

In a three-part investigation, we developed, trialed, and compared WCT differentiation models comprised of newly engineered and previously described features derived from WCT and baseline ECG data. In Part 1, we trained and tested different ML models using features derived from computerized ECG measurements present on the WCT ECG alone. Herein (in Part 1), we evaluate engineered features relating to the orientation of QRS complex polarity in all ECG leads during the WCT itself (i.e., WCT-PC). In Part 2, we trained and tested ML models using features that may be formulated from paired WCT and baseline ECGs. Herein (in Part 2), we evaluate features relating to the presence or absence of QRS polarity shifts between the WCT and baseline ECG (i.e., QRS-PS). In Part 3, we trained and tested ML models that incorporate all features used in Parts 1 and 2. The overarching structure and methodology of our study are visually detailed in Supplemental Fig. S1.

### ECG selection

All paired WCT and baseline ECGs were recorded within clinical settings. ECGs were standard 12-lead recordings (paper speed: 25 mm/s and voltage calibration: 10 mm/mV) accessed from data archives provided by a proprietary ECG interpretation software system (MUSE [GE Healthcare; Milwaukee, WI]). WCTs were required to satisfy standard WCT criteria (QRS duration ≥ 120 ms and ventricular rate ≥ 100 beats per minute) and possess an official ECG interpretation of (i) 'ventricular tachycardia', (ii) 'supraventricular tachycardia', or (iii) 'wide complex tachycardia'. Baseline ECGs were defined as the (i) first non-WCT rhythm recorded *after* the WCT event (training cohort [Institution #1]) or (ii) the most proximate non-WCT rhythm to the WCT event (Testing cohort [Institution #2]).

Polymorphic WCTs and WCTs demonstrating grossly irregular atrioventricular conduction (e.g., atrial fibrillation or atrial flutter with variable atrioventricular block) were excluded, and ECGs demonstrating truncated WCTs (e.g., a brief run of non-sustained VT) occurring within a dominant baseline heart rhythm (e.g., normal sinus rhythm) were not evaluated. If a WCT did not have a baseline ECG or definitive clinical diagnosis established by the patient's overseeing physician, it was excluded from further analysis. Among patients with multiple WCT events, any subsequent WCT ECGs occurring after the first event were excluded, ensuring that each selected patient had only one pair of WCT and baseline ECGs for analysis.

### Heart rhythm diagnoses

Heart rhythm diagnoses (i.e., VT or SWCT) were established by the patient's supervising physician. Heart rhythm diagnoses were organized according to whether they were supported by a corroborating electrophysiology procedure (EP) or implantable intracardiac device recordings (i.e., gold standard cohort vs. non-gold standard cohort).

### Reporting summary

Further information on research design is available in the Nature Portfolio Reporting Summary linked to this article.

### Study cohorts

#### Training cohort—Institution #1

The Training cohort comprised 421 consecutive patients with paired WCT and baseline ECGs acquired at the Mayo Clinic Rochester or Mayo Clinic Health System of South Eastern Minnesota (September 1st, 2011 through November 30th, 2016). Of the 421 patients, 192 heart rhythm diagnoses (i.e., VT or SWCT) were established *with* corroborating EP or implantable intracardiac device recordings (i.e., gold standard cohort). Of the 421 patients, 229 heart rhythm diagnoses were established *without* corroborating EP or implantable intracardiac device recordings. The ECG selection processes and clinical characteristics of this patient cohort are thoroughly described in previous reports[19,20,25,26]. The Mayo Clinic Institutional Review Board approved patient data acquisition and analysis. Informed consent from subjects was waived due to the retrospective nature of this study and the minimal risk it posed to patients.

#### Testing cohort—Institution #2

The testing cohort comprised 235 consecutive patients with paired WCT and baseline ECGs obtained at Barnes-Jewish Hospital in St. Louis (January 1st, 2012 through December 31st, 2014). Supplemental Fig. S2 shows a flow diagram of testing cohort selection. Of the 235 patients, 103 heart rhythm diagnoses (i.e., VT or SWCT) were established *with* corroborating EP or implantable intracardiac device recordings (i.e., gold standard cohort). Of the 235 patients, 132 heart rhythm diagnoses were established *without* a corroborating EP or implantable intracardiac device recordings (i.e., non-gold standard cohort). Patient data acquisition and analysis was approved by the Human Research Protection Office of Washington University in St. Louis, Missouri. Informed consent from subjects was waived due to the retrospective nature of this study and the minimal risk it posed to patients.

### Computerized ECG measurements

Standard computerized ECG measurements for WCT and baseline ECGs were automatically generated by *GE Healthcare's* MUSE ECG interpretation software. Computerized QRS amplitude (μV) and TVA (time-voltage area [μV·ms]) measurements of waveforms above (*r/R* and *r'/R'*) and below (*q/QS, s/S,* and *s'/S'*) the isoelectric baseline were automatically derived from the dominant QRS complex template within each lead of the 12-lead ECG (Supplemental Fig. S3). Only amplitude and TVA measurements representative of QRS complex waveforms were analyzed. Measurements of the ventricular pacing stimuli or ECG artifact were excluded from our analysis.

### WCT differentiation features

#### WCT QRS duration (ms)

QRS duration of the WCT was automatically generated by *GE Healthcare's* MUSE ECG interpretation software package.

#### Percent monophasic time–voltage area (%)

PMonoTVA is the percentage (%) of QRS TVA (time-voltage area) contained by monophasic QRS complexes on the 12-lead ECG[22]. This parameter is calculated as the percentage ratio of the sum of QRS TVA from ECG leads with monophasic QRS complexes (i.e., monophasic TVA) to the entire sum of QRS TVA from all ECG leads (i.e., monophasic TVA plus multiphasic TVA) (Supplemental Fig. S4). Representative TVA measurements are attained from QRS complex waveforms (*q/QS, r/R, s/S, r'/R',* and

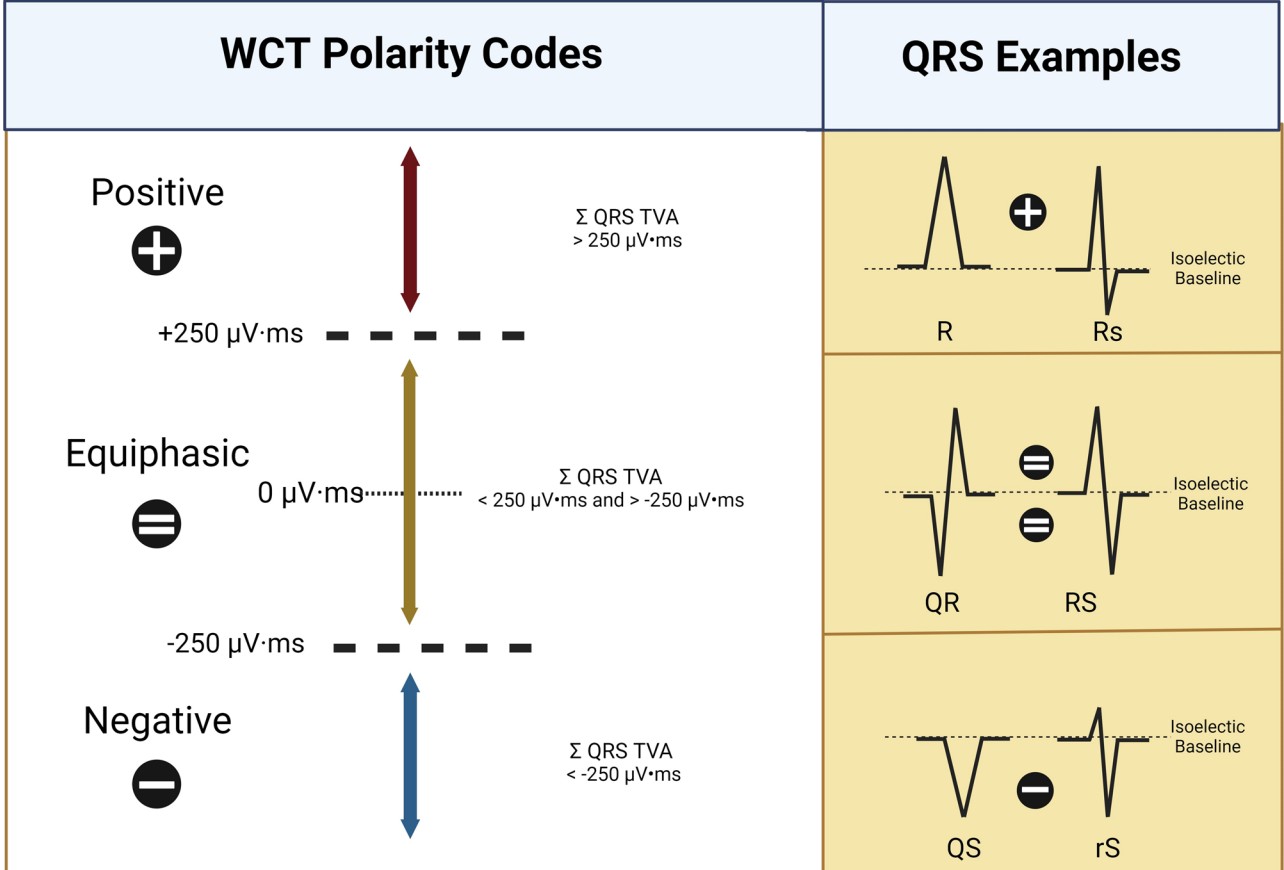

**Fig. 1 | WCT Polarity Codes were determined by way of summating QRS complex waveform TVA (µV·ms) above (r/R and r'/R') and below (q/QS, s/S, and s'/S') the isoelectric baseline.** For each calculation, the amplitude (µV ms) of QRS complex waveforms having a downward orientation (i.e., q/QS, s/S, and s'/S') were considered numerically negative, while the TVA (µV ms) of QRS complex waveforms having an upright orientation (i.e., r/R and r'/R') were considered numerically positive. If the sum of (Σ) QRS complex waveforms (i.e., q/QS + r/R + s/S + r'/R' + s'/S') was greater than 250 µV ms above the isoelectric baseline (i.e., Σ QRS TVA > 250 µV ms), the QRS complex is defined as having positive (+) polarity. If the sum of QRS complex waveforms was more than 250 µV ms beneath the isoelectric baseline (i.e., Σ QRS TVA < −250 µV ms), the QRS complex is defined as having negative (−) polarity. If the sum of QRS complex waveforms falls between 250 µV ms above or beneath the isoelectric baseline (i.e., 250 µV ms ← Σ QRS TVA → −250 µV ms), the QRS complex is defined as having equiphasic (=) polarity. *Created with BioRender.com.*

s'/S') of the dominant QRS complex template of individual leads of the 12-lead ECG.

### WCT polarity code (WCT-PC)

The term WCT-PC denotes the polarity (or direction) of the QRS complex (i.e., positive [+], negative [−], or equiphasic [=]) for individual leads of the recorded WCT. Specifically, QRS complex polarity refers to the dominant direction (i.e., upright, downward, or equiphasic) to which individual QRS complexes of the 12-lead ECG are oriented (Fig. 1).

For this analysis, positive (+), negative (−), and equiphasic (=) QRS polarity was determined by the sum (Σ) of QRS complex waveform TVAs (µV ms) above (r/R and r'/R') and below (q/QS, s/S, and s'/S') the isoelectric baseline. In other words, QRS polarity is determined by the cumulative summation of all QRS waveform integrals, which corresponds to the area under the QRS waveforms. For calculation purposes, the TVA of QRS complex waveforms having a downward orientation (i.e., q/QS, s/S, and s'/S') were considered numerically negative, while the TVA of QRS complex waveforms having an upright orientation (i.e., r/R and r'/R') were considered numerically positive. If the sum of QRS complex waveforms (i.e., q/QS + r/R + s/S + r'/R' + s'/S') was greater than 250 µV ms above the isoelectric baseline (i.e., Σ QRS TVA > 250 µV ms), the QRS complex is defined as having positive (+) polarity. If the sum of QRS complex waveforms was more than 250 µV ms beneath the isoelectric baseline (i.e., Σ QRS TVA < −250 µV ms), the QRS complex is defined as having negative (−) polarity. If the sum of the QRS

complex waveforms falls within the range of 250 µV ms above or below the isoelectric baseline (i.e., −250 µV ms ≤ Σ QRS TVA ≤ 250 µV ms), the QRS complex is defined as having equiphasic (=) polarity.

### QRS polarity shift (QRS-PS)

The term QRS-PS will be employed to characterize changes, whether they occur or not, in QRS polarity between a baseline and a WCT rhythm for individual ECG leads. Characterization of QRS polarity changes, or lack thereof, can be organized according to the type of change between a baseline and WCT rhythm (Fig. 2). In specific terms, polarity changes between a WCT and baseline ECG may be characterized in one of 9 ways (WCT ECG → Baseline ECG or vice versa): (i) equiphasic (=) → equiphasic (=), (ii) positive (+) → positive (+), (iii) negative (−) → negative (−), (iv) positive (+) → negative (−), (v) positive (+) → equiphasic (=), (vi) negative (−) → positive (+), (vii) negative (−) → equiphasic (=), (viii) equiphasic (=) → positive (+), and (ix) equiphasic (=) → negative (−). In general terms, polarity changes between a WCT and baseline ECG may be characterized in one of three ways: (i) polarity shift, (ii) partial polarity shift, or (iii) no polarity shift.

For this analysis, polarity shift is defined by the occurrence of a change in QRS complex polarity between the WCT and baseline ECG. For a polarity shift to occur, individual QRS complexes of the 12-lead ECG having a dominate QRS polarity (positive [+] or negative [−]) on the baseline or WCT ECG must transform into the opposite dominate QRS polarity on the

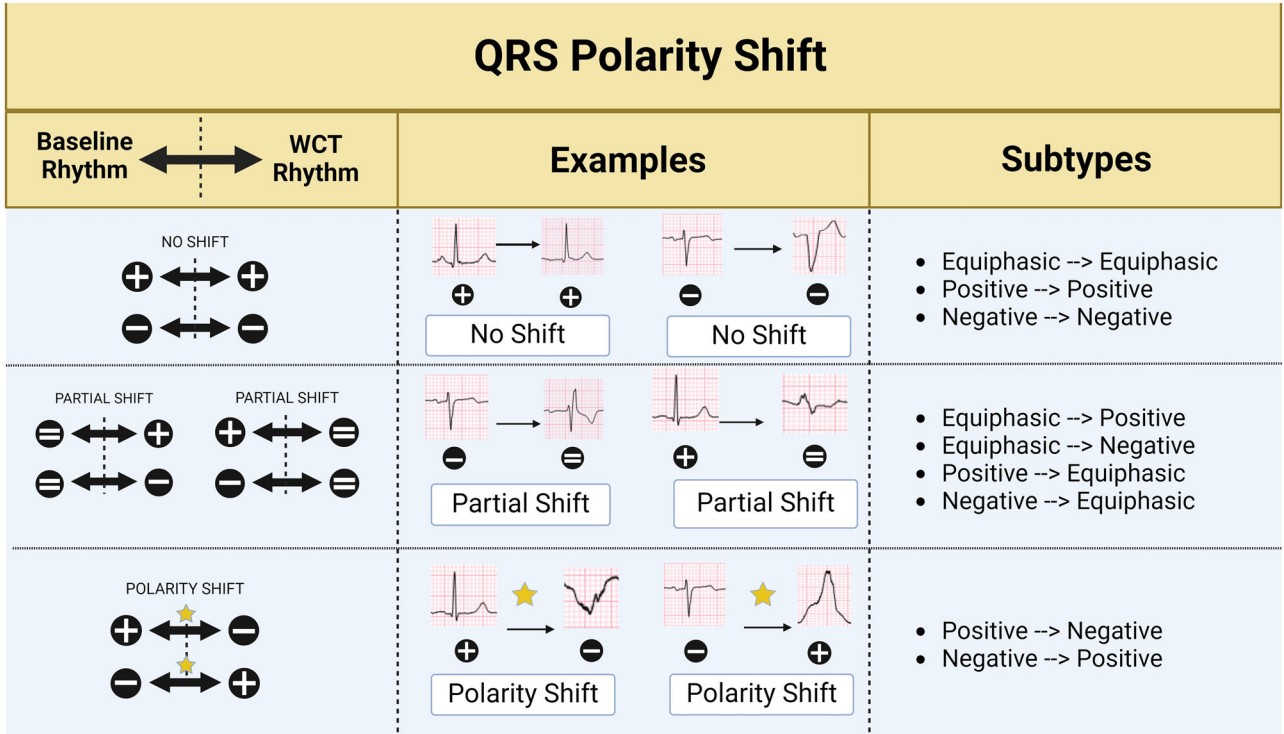

**Fig. 2 | Examples of QRS polarity shift possibilities during the transition of a baseline ECG into a WCT ECG, and vice versa.** Specified subtypes for QRS-polarity shift are shown: ([i] no polarity shift, [ii] polarity shift, and [iii] partial polarity shift) and nine more specific subtypes ([i] equiphasic (=) → equiphasic (=), [ii] positive (+) → positive (+), [iii] negative (−) → negative (−), [iv] positive (+) → negative (−), [v] positive (+) → equiphasic (=), [vi] negative (−) → positive (+), [vii] negative (−) → equiphasic (=), [viii] equiphasic (=) → positive (+), and [ix] equiphasic (=) → negative (−)). *Created with BioRender.com.*

corresponding WCT or baseline ECG, respectively. For example, a polarity shift occurs if an individual ECG lead demonstrates positive (+) QRS complex polarity during the baseline heart rhythm that later transforms into a negative (−) QRS complex polarity during a WCT. Similarly, a polarity shift is also present when an ECG lead demonstrates negative (−) QRS complex polarity during the baseline heart rhythm but a positive (+) QRS complex polarity during a WCT. Partial polarity shift is defined by QRS morphology transformation into or out of an equiphasic (=) QRS complex. For a partial polarity shift to occur, individual QRS complexes of baseline or WCT ECG must either transform into or out of an equiphasic (=) QRS complexes on the corresponding baseline or WCT ECG. No polarity shift is defined by the absence of QRS polarity change for individual leads between the baseline and WCT ECG. For example, no polarity shift is present when an individual ECG lead demonstrates positive (+) QRS complex polarity during the baseline heart rhythm and WCT. Similarly, no polarity shift is present for an individual ECG lead demonstrating a negative (−) QRS complex or equiphasic polarity for both the baseline and WCT ECGs. Supplementary Fig. S5 offers examples of polarity shift, partial polarity shift, and no polarity shift for ECG leads of the frontal and horizontal plane.

## Statistics and reproducibility
In Part 1, five ML modeling techniques (logistic regression [LR], artificial neural network [ANN], Random Forests [RF], support vector machine [SVM], and ensemble learning [EL]) were used to train and test binary classification models that may be implemented on the WCT ECG alone, with each model incorporating fourteen covariates (i.e., features): WCT QRS duration (ms), PMonoTVA (%), and WCT-PC$_{ECG\ lead\ X}$ (X12 covariates in total [one per ECG lead]). For any given ECG lead, WCT-PC had three potential categorical values: (i) positive (+), (ii) negative (−), or (iii) equiphasic (=).

In Part 2, the five ML modeling techniques were used to train and test binary classification models that may be implemented on paired WCT and baseline ECGs, with each model incorporating fourteen covariates: WCT

QRS duration (ms), PMonoTVA (%), and QRS-PS$_{ECG\ lead\ X}$ (X12 covariates in total [one per ECG lead]). For any given ECG lead, QRS-PS covariates had three potential categorical options: (i) polarity shift, (ii) partial polarity shift, or (iii) no polarity shift.

In Part 3, the five ML modeling techniques were used to train and test binary classification models incorporating twenty-six covariates: WCT QRS duration (ms), PMonoTVA (%), WCT-PC$_{ECG\ lead\ X}$ (X12 covariates in total [one per ECG lead]), and QRS-PS$_{ECG\ lead\ X}$ (X12 covariates in total [one per ECG lead]). For any given ECG lead, WCT-PC had three potential categorical values: (i) positive (+), (ii) negative (−), or (iii) equiphasic (=). Similarly, for any given ECG lead, QRS-PS covariates had 3 potential categorical options: (i) polarity shift, (ii) partial polarity shift, or (iii) no polarity shift.

Categorical variables were compared using Chi-square tests. Wilcoxon rank-sum tests were used to compare continuous variables. Positive likelihood ratios (+LR) were used to evaluate the individual discriminatory capacity of WCT-PC$_{ECG\ lead\ X}$ and QRS-PS$_{ECG\ lead\ X}$ for the correct rhythm diagnosis. Outlier values for each parameter were winsorized to diminish undue influence on model coefficients. Heart rhythm classification (i.e., VT or SWCT) by each model was established using a pre-specified VT probability partition of 50% (i.e., VT ≥ 50% and SWCT < 50%). Performance metrics (i.e., accuracy, sensitivity, specificity, and AUC) for each model were assessed according to their agreement with the correct heart rhythm diagnosis. A comparison of fit between ML models was completed using a Delong test. All statistical analysis was performed using R Statistical Software (R Foundation for Statistical Computing, Vienna, Austria).

## Results
### Cohort characteristics
**Training cohort—Institution #1.** Clinical characteristics of the training cohort are shown in Supplementary Table S1. Among training cohort patients, the VT group included more patients with coronary artery disease, prior myocardial infarction, ongoing antiarrhythmic drug use,

**Fig. 3 | Forest plots demonstrating the discriminatory capacity of positive (+) and negative (−) WCT-PCs among individual ECG leads.** Plot squares and intervals represent the mean and confidence intervals, respectively (blue Institution #1, red Institution #2). Upper panel: plot demonstrating positive (+) likelihood ratios among ECG leads demonstrating a positive (+) WCT-PC, as compared to negative (−) or equiphasic (=) WCT-PCs. Lower panel: plot demonstrating positive (+) likelihood ratios among ECG leads demonstrating a negative (−) WCT-PC, as compared to positive (+) or equiphasic (=) WCT-PCs. Abbreviations: WCT-PC WCT polarity code.

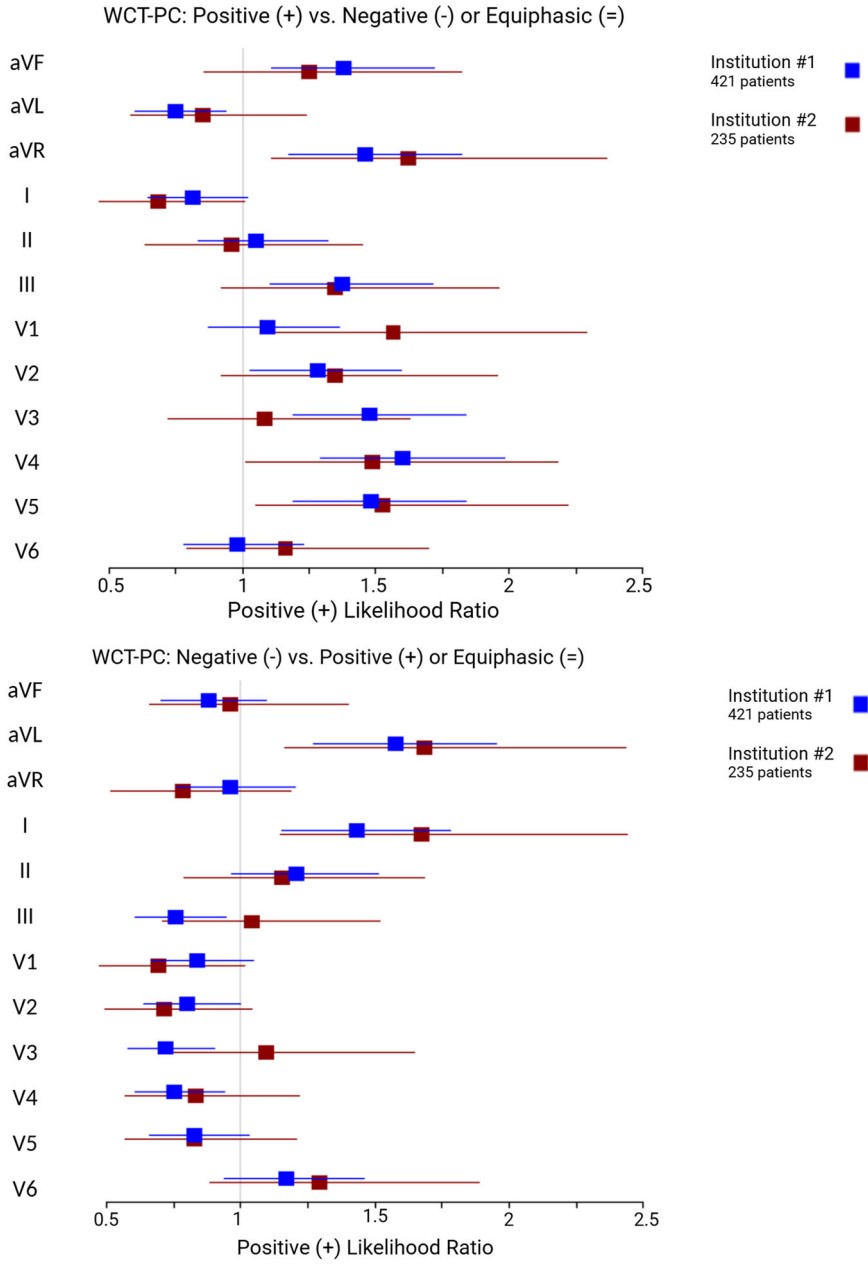

ischemic cardiomyopathy, and an implantable cardioverter-defibrillator (ICD). The SWCT group included more patients with an implanted pacemaker. The VT group comprised more patients with a severely depressed (≤30%) left ventricular ejection fraction (LVEF), whereas the SWCT group included more patients with a preserved (≥50%) LVEF. A majority of patients with VT diagnoses had a corroborating EP or intracardiac device recording. Conversely, a minority of SWCT patients had a corroborating EP or intracardiac device recording. The median time between the WCT and baseline ECG was 5.6 h (interquartile range: 1.0, 41.8).

**Testing cohort—Institution #2.** Clinical characteristics of the testing cohort are shown in Supplementary Table S2. Among testing cohort patients, the VT group included more patients with ongoing antiarrhythmic drug use, ischemic cardiomyopathy, and an ICD. Similar to the training cohort, the VT group comprised more patients with a severely depressed (≤30%) LVEF, while the SWCT group included more patients with a preserved (≥50%) LVEF. Similar to the training cohort, most VT patients, and a minority of SWCT patients, had a corroborating EP or

intracardiac device recording. The median time between the WCT and baseline ECG was 16.2 h (interquartile range: 1.9, 67.5).

**WCT polarity code (WCT-PC)**

The frequency of various types of WCT-PCs for VT and SWCT groups within the testing cohort is shown in Supplemental Table S3. The discriminatory capacity of positive (+) and negative (−) QRS polarity among individual ECG leads of the WCT is illustrated in Fig. 3.

For the frontal ECG plane, positive (+) QRS polarity in lead aVR modestly favored VT in both the training and testing cohorts. Positive (+) QRS polarity in lead aVF and III appeared to marginally favor VT in the training cohort only. Negative (−) QRS polarity complexes in leads aVL and I modestly favored VT in both the training and testing cohorts. Conversely, a negative (−) QRS polarity in lead III slightly favored SWCT. Otherwise, QRS complex polarity patterns in the frontal ECG plane did not significantly favor VT or SWCT.

For the horizontal ECG plane, positive (+) QRS polarity in V4 and V5 marginally favored VT in the training and testing cohorts. Positive (+) QRS polarity in leads V2 and V3 slightly favored VT in the training cohort only;

**Fig. 4 | Forest plots demonstrating the discriminatory capacity QRS-PS among individual ECG leads.** Plot squares and intervals represent the mean and confidence intervals, respectively (blue Institution #1, red Institution #2). Upper panel: plot demonstrating positive (+) likelihood ratios among ECG leads demonstrating a polarity shift or partial polarity shift, as compared to no polarity shift. Lower panel: plot demonstrating negative (−) likelihood ratios among ECG leads demonstrating a polarity shift or partial polarity shift, as compared to no polarity shift. Abbreviations: QRS-PS QRS polarity shift.

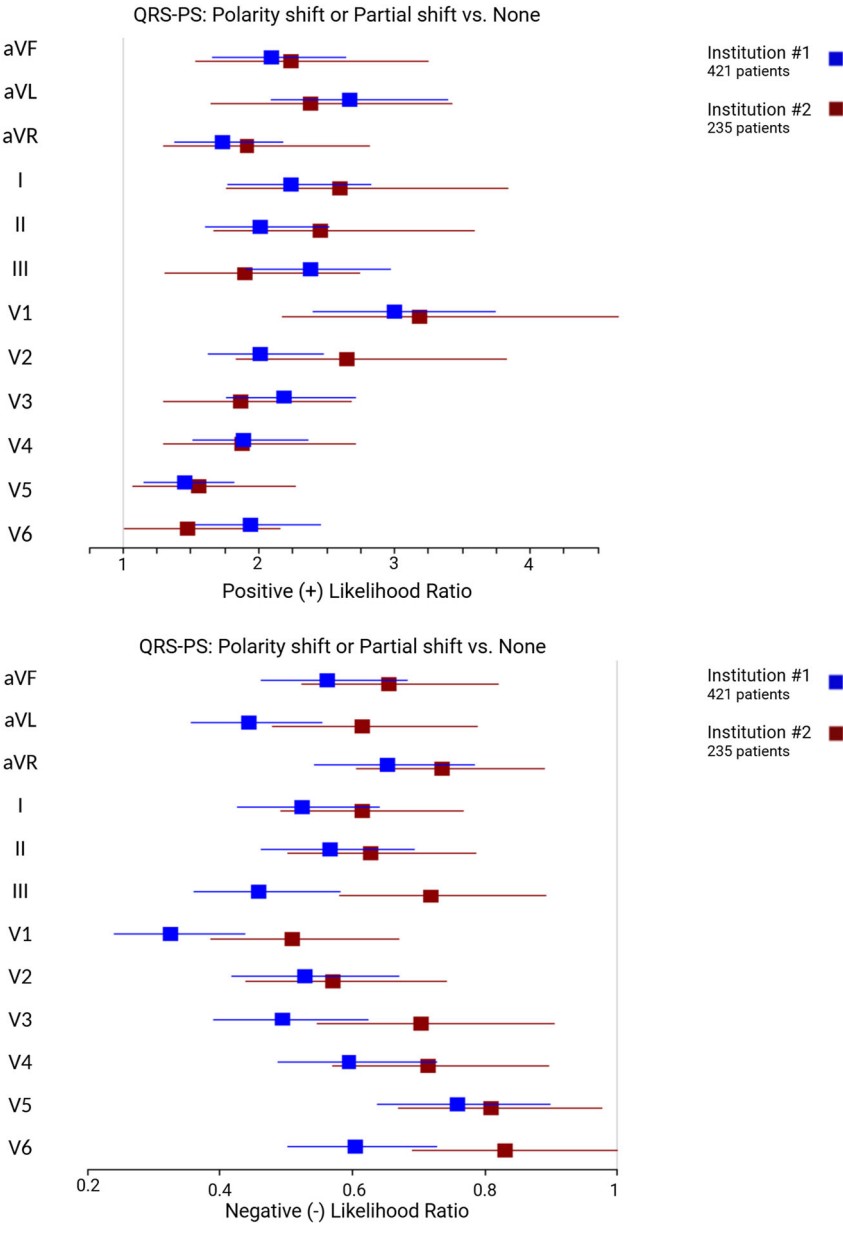

but in lead V1, positive (+) QRS polarity slightly favored VT in the testing cohort. Negative (−) QRS polarity complexes slightly favored SWCT in leads V3 and V4 in the training cohort only. Otherwise, QRS complex polarity patterns in the horizontal ECG plane did not significantly favor VT or SWCT.

### QRS polarity shift (QRS-PS)

The frequency of various types of QRS-PSs for VT and SWCT groups within the testing cohort is shown in Supplemental Table S4 and Supplementary Data File D1. The discriminatory capacity any polarity shift occurring among leads of the 12-lead ECG are illustrated in Fig. 4.

For all ECG leads the presence of a polarity shift strongly favored VT for both the training and testing cohorts. Conversely, for all ECG leads the absence of a polarity shift strongly favored SWCT in both the training and testing cohorts.

### Part 1: ML models—WCT ECG data alone

Diagnostic performance metrics of the five ML models in the training cohort are summarized in Supplemental Table S5. Table 1 and Fig. 5 demonstrate the diagnostic performance of various modeling subtypes as applied to

subjects with a gold standard and non-gold standard diagnosis in the testing cohort, respectively. Comparisons of ML model performance are shown in Supplemental Table S6. In the gold standard cohort, ML model techniques demonstrated similar diagnostic performance (AUC range: 0.86 to 0.88). In the non-gold standard cohort, ML model techniques demonstrated similar diagnostic performance (AUC range: 0.86 to 0.87). Specific tuning parameters used for ML model training are shown in Supplemental Table S7. The importance of various WCT differentiation features of the RF model is shown in Supplemental Table S8.

### Part 2: ML models—paired WCT and baseline ECG data

Diagnostic performance metrics of the five ML models, when applied to the training cohort, are summarized in Supplemental Table S9. Table 2 and Fig. 6 demonstrate the diagnostic performance of the ML modeling subtypes when applied to subjects with a gold standard and non-gold standard diagnosis in the testing cohort, respectively. Comparisons of ML model performance are shown in Supplemental Table S10. Among subjects with a gold standard diagnosis, ML model techniques demonstrated similar diagnostic performance (AUC range: 0.90 to 0.93). Among subjects with a non-gold standard diagnosis, ML model techniques demonstrated similar

**Table 1 | Study Part 1: ML model implementation on testing cohort**

|  | Cohort | Accuracy (%) | Sensitivity (%) | Specificity (%) | AUC |
|---|---|---|---|---|---|
| Logistic regression | Gold standard | 78.6 | 72.9 | 83.6 | 0.858 |
|  | Non-gold standard | 84.8 | 73.1 | 87.7 | 0.857 |
| Artificial neural network | Gold standard | 78.6 | 72.9 | 83.6 | 0.865 |
|  | Non-gold standard | 84.8 | 69.2 | 88.7 | 0.866 |
| Random forest | Gold standard | 77.7 | 70.8 | 83.6 | 0.856 |
|  | Non-gold standard | 81.0 | 61.5 | 85.8 | 0.871 |
| Support vector machine | Gold standard | 80.6 | 77.1 | 83.6 | 0.876 |
|  | Non-gold standard | 82.5 | 69.2 | 85.8 | 0.849 |
| Ensemble learner | Gold standard | 80.6 | 75.0 | 85.5 | 0.873 |
|  | Non-gold standard | 84.8 | 69.2 | 88.7 | 0.870 |

Summary of the diagnostic performance of various ML model subtypes in Part 1 (using 14 features) on the gold standard and non-gold standard cohorts of the testing cohort (from Institution #2).
Abbreviations: *ML* machine learning.

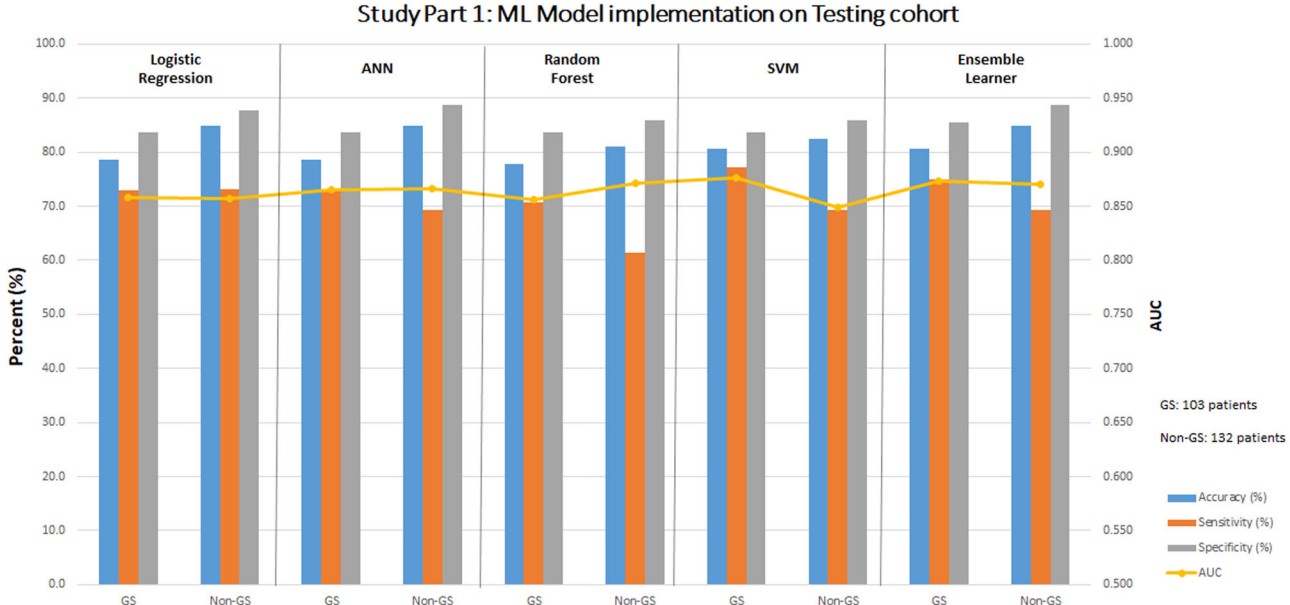

**Fig. 5 | Study Part 1: diagnostic performance metrics of ML Models applied on the testing cohort.** Abbreviations: ANN artificial neural network, AUC area under the receiver operating curve, GS gold standard, non-GS non-gold standard, ML machine learning, SVM support vector network.

diagnostic performance (AUC range: 0.91 to 0.92). Specific tuning parameters used for ML model training are shown in Supplemental Table S11. The importance of various WCT differentiation features of the RF model is shown in Supplemental Table S12.

### Part 3: ML models—WCT ECG data + paired WCT and baseline ECG data
Diagnostic performance metrics of the five ML models, when applied to the training cohort, are summarized in Supplemental Table S13. Table 3 and Fig. 7 demonstrate the diagnostic performance of the ML modeling subtypes when applied to subjects with a gold standard and non-gold standard diagnosis in the testing cohort, respectively. Comparisons of ML model performance are shown in Supplemental Table S14. Specific tuning parameters used for ML model training are shown in Supplemental Table S15. The importance of various WCT differentiation features in the RF model is shown in Supplemental Table S16.

In Part 3, within the gold standard cohort, ML models displayed varying capabilities in diagnostic performance, with AUC values ranging

from 0.72 to 0.93. Notably, SVM and RF methods outperformed other approaches, while the LR method was the weakest performer. Specific to the LR model, all covariates had Variance Inflation Factor values greater than 5, indicating substantial multicollinearity. In the RF model, six parameters were identified as non-important factors (importance score 0.00) for model outputs.

### Discussion
In this study, we have outlined and examined innovative diagnostic parameters that can be integrated into automated ML models aimed at achieving precise differentiation of WCTs. More specifically, we have developed new automated algorithms for WCT differentiation that utilize (i) the direction of QRS complex polarity in WCTs, referred to as the WCT-PC (WCT Polarity Code), and (ii) the changes in QRS polarity direction between the WCT and baseline ECG, termed QRS-PS (QRS polarity shift).

We trained and tested models using computerized ECG data from (i) WCT ECG alone (Part 1) and (ii) paired WCT and baseline ECGs (Parts 2

## Table 2 | Study Part 2: ML model implementation on testing cohort

|  | Cohort | Accuracy (%) | Sensitivity (%) | Specificity (%) | AUC |
|---|---|---|---|---|---|
| Logistic regression | Gold standard | 80.6 | 70.8 | 89.1 | 0.900 |
|  | Non-gold standard | 89.4 | 80.8 | 91.5 | 0.910 |
| Artificial neural network | Gold standard | 81.6 | 77.1 | 85.5 | 0.901 |
|  | Non-gold standard | 87.1 | 80.8 | 88.7 | 0.907 |
| Random forest | Gold standard | 85.4 | 77.1 | 92.7 | 0.927 |
|  | Non-gold standard | 90.2 | 84.6 | 91.5 | 0.913 |
| Support vector machine | Gold standard | 80.6 | 72.9 | 87.3 | 0.921 |
|  | Non-gold standard | 88.6 | 84.6 | 89.6 | 0.907 |
| Ensemble learner | Gold standard | 88.6 | 84.6 | 89.6 | 0.911 |
|  | Non-gold standard | 82.5 | 75.0 | 89.1 | 0.920 |

Summary of the diagnostic performance of various ML model subtypes in Part 2 (using 14 features) on the gold standard and non-gold standard cohorts of the testing cohort (from Institution #2).
Abbreviations: *ML* machine learning.

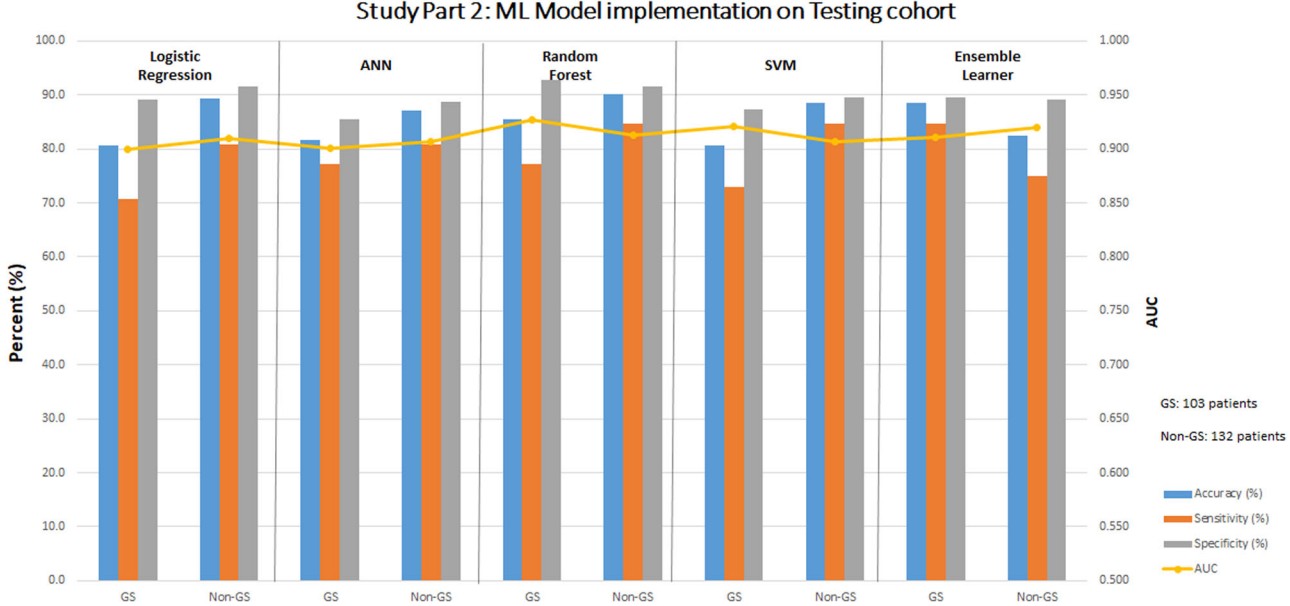

**Fig. 6 | Study Part 2: diagnostic performance metrics of ML Models applied on the testing cohort.** Abbreviations: ANN artificial neural network, AUC area under the receiver operating curve, GS gold standard, non-GS non-gold standard, ML machine learning, SVM support vector network.

## Table 3 | Study Part 3: ML model implementation on testing cohort

|  | Cohort | Accuracy (%) | Sensitivity (%) | Specificity (%) | AUC |
|---|---|---|---|---|---|
| Logistic regression | Gold standard | 72.8 | 66.7 | 78.2 | 0.724 |
|  | Non-gold standard | 82.6 | 73.1 | 84.9 | 0.790 |
| Artificial neural network | Gold standard | 76.7 | 68.8 | 83.6 | 0.888 |
|  | Non-gold standard | 86.4 | 80.8 | 87.7 | 0.882 |
| Random forest | Gold standard | 86.4 | 83.3 | 89.1 | 0.921 |
|  | Non-gold standard | 90.2 | 84.6 | 91.5 | 0.928 |
| Support vector machine | Gold standard | 82.5 | 81.3 | 83.6 | 0.925 |
|  | Non-gold standard | 89.4 | 84.6 | 90.6 | 0.890 |
| Ensemble learner | Gold standard | 76.7 | 68.8 | 83.6 | 0.894 |
|  | Non-gold standard | 85.6 | 76.9 | 87.7 | 0.884 |

Summary of the diagnostic performance of various ML model subtypes in Part 3 (using 26 features) on the gold standard and non-gold standard cohorts of the testing cohort (from Institution #2).
Abbreviations: *ML* machine learning.

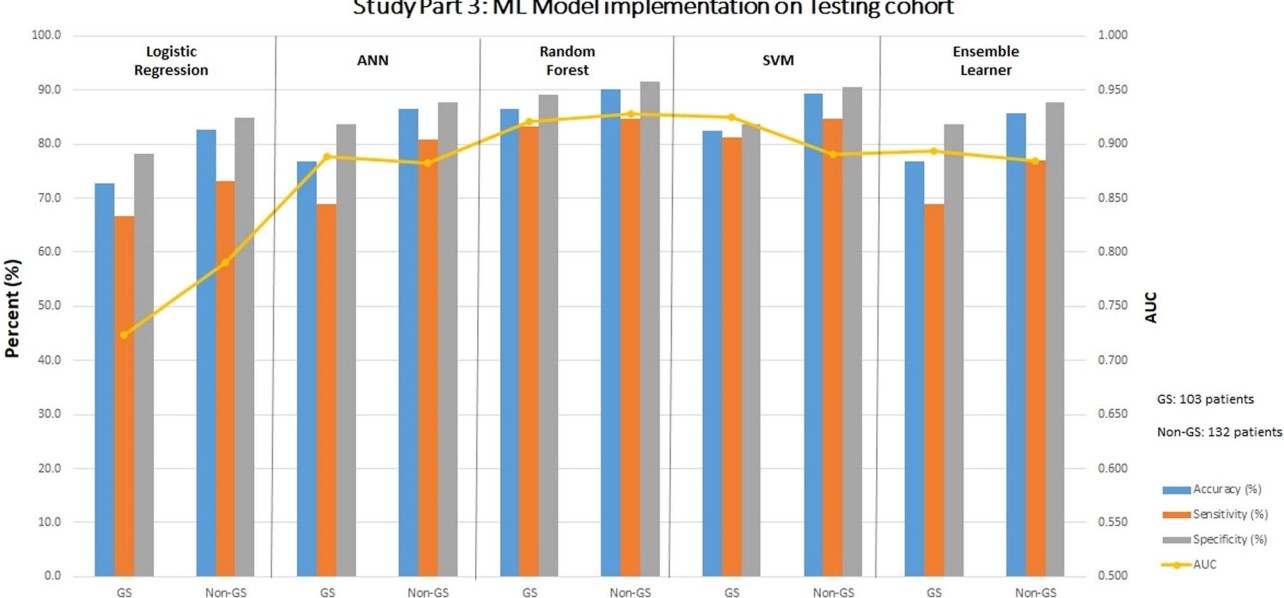

**Fig. 7 | Study Part 3: diagnostic performance metrics of ML Models applied on the testing cohort.** Abbreviations: ANN artificial neural network, AUC area under the receiver operating curve, GS gold standard, non-GS non-gold standard, ML machine learning, SVM support vector network.

and 3). In Part 1, we observed that the incorporation of features derived from the WCT data alone (including WCT-PCs) into various ML modeling techniques resulted in favorable diagnostic performance. In Parts 2 and 3, we found that features derived from paired WCT and baseline ECG data (i.e., QRS-PS) resulted in improved diagnostic performance. However, the aggregation of all available features used in Parts 1 and 2 in Part 3 did not lead to a meaningful improvement in diagnostic performance. This finding indicates a ceiling in diagnostic performance due to parameter multicollinearity. Lastly, we observed that effective ML model performance was maintained irrespective of its implementation on patients who have or do not have a corroborating gold standard diagnosis.

We investigated whether WCT-PCs (i.e., positive [+], negative [−], or equiphasic [=]) of individual leads of the WCT itself can be leveraged to arrive an accurate VT or SWCT diagnosis. We hypothesized VT and SWCT rhythms would, in many cases, demonstrate a unique 'WCT-PC signature' that would enable accurate WCT differentiation using ML techniques. For example, WCT rhythms with a 'northwest axis' ([+] QRS polarity in lead aVR) and a 'QS pattern' in lead V6 ([−] QRS polarity in V6) are more likely to be VT, especially since this WCT-PC signature would not be expected among patients SWCT due to aberrancy. Similarly, we would expect WCT rhythms with a rightward axis ([−] QRS polarity in lead I) and left bundle branch block (LBBB) pattern ([−] QRS polarity in V1) would be more consistent with VT than SWCT. Prior authors[6,8] have identified both aforementioned examples as patterns as being more consistent with VT. Therefore, we sought to determine whether potential QRS-PC signatures (known or unknown) could be used to differentiate VT and SWCT. Therefore, we incorporated WCT-PCs into various ML modeling techniques and methods capable of assimilating features with disparate non-parametric relationships for the purpose of VT and SWCT classification.

After assigning positive (+), negative (−), or equiphasic (=) labels for all QRS complexes for individual leads of the recorded WCT, we detected relationships between QRS complex polarity and the underlying WCT diagnosis. For example, we observed that a positive (+) QRS polarity in lead aVR favored VT in both the training and testing cohorts. Similarly, we observed that a negative (−) QRS polarity in lead I favored VT in both the training and testing cohorts. Nonetheless, among the few individual ECG leads that did demonstrate a relationship between QRS complex polarity

and the underlying WCT diagnosis, the influence of QRS-PC was objectively small. However, upon incorporating WCT-PCs of all leads into various ML modeling techniques, we observed that models leveraging QRS-PCs efficiently differentiate WCTs. In this case, ML methods appear to be able to decipher complex non-parametric relationships between WCT-PCs to achieve a strong overall diagnostic performance.

We evaluated whether the characterization QRS-PSs between the WCT and baseline ECG could be leveraged in discriminating VT and SWCT. We hypothesized that the presence of any polarity shift (positive [+] QRS polarity transforming into negative [−] QRS polarity, or vice versa) between the baseline and WCT ECGs would be highly predictive of VT. Conversely, we expected that the absence of a polarity shift would be more consistent with SWCT. The electrophysiological basis for our hypotheses relates to the differences by which VT and SWCT ordinarily depolarize the ventricular myocardium. In the case of SWCT, it is quite common for the means of ventricular depolarization to be the exact same or very similar to that of the baseline heart rhythm (e.g., normal sinus rhythm). As such, it is unusual for a polarity shift to occur among SWCTs; rather, SWCTs would more commonly demonstrate a lack of any polarity shift. On the other hand, VT is known to demonstrate substantial 'electrical freedom' compared to its relatively constrained SWCT counterpart. Considering that VT can originate and spread from any part of the left or right ventricles, it can produce ventricular depolarization wavefronts that move in the opposite direction of the baseline heart rhythm, essentially demonstrating a complete reversal of the mean electrical vector orientation. As such, many VT rhythms would be expected to elicit a polarity shift between the baseline and WCT rhythm. This concept[27] was similarly leveraged by other recently described features, (e.g., frontal and horizontal PAC) that were incorporated into WCT differentiation algorithms[19–23].

In this analysis, we observed that any polarity shift is highly predictive of VT. Conversely, we observed that the absence of polarity shift is highly predictive of SWCT. Upon incorporating QRS-PS covariates from all twelve ECG leads into various ML modeling techniques, we observed a robust capacity for accurately differentiating VT and SWCT. In both Parts 2 and 3 of this analysis, ML models yielded strong performance using QRS-PS covariates. Moreover, we observed QRS-PS covariates to have uniformly stronger influence than WCT-PC covariates, as demonstrated by RF importance scores.

Ideally, accurate and reliable discrimination of VT and SWCT would occur automatically upon 12-lead ECG acquisition. Unfortunately, CEI (computerized ECG interpretation) software programs have not yet achieved sufficient diagnostic accuracy for the interpretation of many complex heart rhythms[28], including WCTs. At present, contemporary CEI software packages do not reliably differentiate WCTs; rather, instead, most recorded WCT events are given a generic label of 'wide complex tachycardia', which offers little-to-no assistance to clinicians. Consequently, patient-facing clinicians must rely on conventional manual ECG interpretation techniques to accurately and promptly diagnose VT or SWCT. To do this effectively, clinicians must analyze a properly recorded 12-lead ECG that accurately represents the WCT event, and diligently apply the specific electrocardiographic criteria outlined by traditional interpretation methods. Unfortunately, this procedure is commonly thwarted by the improper application or lack of use of manual WCT differentiation methods. As such, the application of manual ECG interpretation algorithms or criteria is unsurprisingly problematic—especially for non-expert clinicians who must promptly diagnose and manage high-acuity patients.

In this work, we developed and trialed means to distinguish VT and SWCT accurately using ECG data provided by the (i) WCT alone and (ii) paired WCT and baseline ECGs. Similar to other recently described methods[19–21,23], we again demonstrate how automated approaches to differentiate WCTs may be developed through the use of available data provided by CEI software. Likewise, we transformed readily available ECG data into mathematically formulated features (e.g., QRS-PS), which may, in turn, be incorporated into models that provide clinicians an impartial binary classification or estimation of VT likelihood (i.e., 0.00% to 99.99% VT probability).

A recognized limitation of recently described automated WCT differentiation methods[19–21] is that they require computerized data provided by *both* the WCT ECG and its corresponding baseline ECG. In Part 1 of this analysis we described means to distinguish VT and SWCT using the WCT alone. Thus irrespective of the presence of a baseline ECG, ML models may be employed. For instance, if a patient does not have a baseline ECG, ML models that use computerized ECG data of the WCT alone can be used. However, if a patient already possesses a digitally archived baseline ECG, or if a new baseline ECG is recorded after the WCT event, VT or SWCT classification may be executed by ML models that leverage paired WCT and baseline ECG comparisons, and thereby supersede the classification from ML models that only use computerized ECG data provided by the WCT itself. Supplementary Fig. S6 demonstrates a proposed framework for the application of WCT differentiation algorithms according to the presence or absence of a baseline ECG.

Ideally, healthcare professionals should be able to combine automatically generated VT probabilities, produced by automated algorithms, with diagnoses derived from traditional WCT differentiation methods, such as the Brugada algorithm[9] or VT score[29]. In a recent analysis[30], we observed that displaying VT probability (or likelihood that the WCT is VT) generated by the VT Prediction Model helped physicians' diagnostic performance in discriminating VT and SWCT. By similar means, commercially available ECG interpretation software platforms that incorporate ML models would be able to help clinicians discriminate VT and SWCT accurately.

This study is best evaluated in the context of its limitations. First, our study examined any patient with a clinically encountered WCT that was formally diagnosed by the patient's overseeing physician. As a result, we evaluated WCTs *with* and *without* a corroborating EP or implanted intracardiac device recording. Although we found that proposed models performed well on the gold standard and non-gold standard cohorts, we acknowledge that a significant proportion of this analysis includes VT or SWCT diagnoses not established by the most robust reference standard. Second, as our analysis examined standard 12-lead ECGs acquired within clinical settings, we cannot make more comprehensive conclusions regarding model performances on various VT and SWCT subtypes (e.g., SWCT due to pre-excitation) delineated by way of an EP. Nevertheless, by

purposely analyzing ECG from 'all comers' who presented with a WCT in genuine clinical circumstances, automated models described in this work would be expected to be more generalizable for broader clinical use. Lastly, the diagnostic performance of automated models was not directly compared with traditional manual WCT differentiation approaches. Recently, our group published a separate analysis comparing the performance of automated models with traditional manual WCT differentiation approaches[31]. We observed automated models demonstrated favorable performance compared to manual WCT differentiation methods across multiple diagnostic metrics.

In conclusion, we herein present automated ML algorithms that leverage features relating to WCT QRS complex polarity (i.e., WCT-PC) and QRS complex polarity shifts between the WCT and baseline ECG (i.e., QRS-PS). By these means, accurate VT and SWCT classification may be accomplished using readily available CEI data provided by the (i) WCT alone or (ii) paired WCT and baseline ECGs.

## Data availability

Source data for Fig. 3 and Fig. 4 for is in Supplementary Data File D2. Source data for Fig. 5, Fig. 6, and Fig. 7 are Table 1, Table 2, and Table 3, respectively.

The complete datasets used in this analysis are not publicly available to protect patient privacy. However, researchers may request access to raw study data for research purposes by contacting the corresponding author, subject to reasonable review and approval.

## Abbreviations list

| | |
|---|---|
| AUC | area under the receiver operating characteristic curve |
| ECG | electrocardiogram |
| EP | electrophysiology procedure |
| ICD | implantable cardioverter-defibrillator |
| LBBB | left bundle branch block |
| LVEF | left ventricular ejection fraction |
| ML | machine learning |
| NPV | negative predictive value |
| PAC | percent amplitude change |
| PMonoTV-A | percent monophasic time-voltage area |
| PPV | positive predictive value |
| PTVAC | percent time-voltage area change |
| QRS-PS | QRS polarity shift |
| RBBB | right bundle branch block |
| SWCT | supraventricular wide complex tachycardia |
| TVA | time-voltage area |
| VT | ventricular tachycardia |
| WCT | wide complex tachycardia |
| WCT-PC | WCT polarity code |

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

## Author contributions

Adam M. May: Conceptualization, data curation, funding acquisition, investigation, methodology, project administration, writing—original draft, writing—review and editing. Bhavesh B. Katbamna: Conceptualization, writing—review, and editing. Preet A. Shaikh: Writing—review and editing. Sarah LoCoco: Data curation, writing—review and editing. Elena Deych: Formal analysis, methodology, writing—review and editing. Ruiwen Zhou: Formal analysis, methodology, writing—review and editing. Lei Liu: Formal analysis, methodology, writing—review and editing. Krasimira M. Mikhova: Writing—review and editing. Rugheed Ghadban: Writing—review and editing. Phillip S. Cuculich: Writing—review and editing. Daniel H. Cooper: Writing—review and editing. Thomas M. Maddox: Writing—review and editing. Peter A. Noseworthy: Writing—review and editing. Anthony H. Kashou: Investigation, methodology, writing—review and editing.

## Competing interests

The authors declare the following competing interests: Adam May and Anthony Kashou are obliged to disclose that they are "would be" beneficiaries of intellectual property that currently has "patent pending" status. The technology in question relates directly to this manuscript's content. This technology was developed without industry funding or influence and was disclosed to the Office of Technology Management/Tech Transfer (OTM) at Washington University in St. Louis, which possesses intellectual property rights. Adam May is an Editorial Board Member for *Communications Medicine* but was not involved in the editorial review or peer review, nor in the decision to publish this article.
