## [Peer review file · Communications Medicine]

Automated Differentiation of Wide QRS Complex Tachycardias Using QRS Complex Polarity

Corresponding Author: Dr Adam May

Version 0:

Reviewer comments:

Reviewer #1

(Remarks to the Author)

The authors of this manuscript present the results of a research that aims to develop automated differentiation of Wide QRS Complex Tachycardia to provide clinicians with estimation of ventricular tachycardia (VT) /supraventricular wide complex tachycardia (SWCT) likelihood. In this study, they described and explored innovative diagnostic characteristics that can be integrated into automated machine learning models directly derived from standard computerized ECG measurements.

They trained and tested the novel machine learning (ML) models using computerized and derived ECG data from WCT ECG alone and paired WCT and baseline ECGs on two separated datasets. The results showed that the incorporation of features derived from the WCT data alone into various ML modeling techniques resulted in favorable diagnostic performance. They also found that novel features derived from paired WCT and baseline ECG data resulted in improved diagnostic performance.

Their conclusion is that accurate VT and SWCT classification may be accomplished using readily available computerized electrocardiogram interpretation data provided by the WCT alone or paired WCT and baseline ECGs.

There are some questions would like to be discussed,

1. There are 72% and 65% of VT cases having Gold Standard Diagnosis in training and testing cohorts, and 26% and 34% of SWCT cases having Gold Standard Diagnosis. Have you thought about balancing the confirmed diagnosis rate in the two group? Such as having diagnosis agreements with experienced rhythm cardiologists.

2. How did you derive the threshold value for the polarity of QRS complex: 250 $\mu\text{V}\cdot\text{ms}$ above or below the isoelectric baseline?

3. The results in Part 3 show adding all parameters (WCT ECG data + Paired WCT and Baseline ECG data) to ML models reduced the diagnosis performance especially for LR method. Have you looked into the correlations among these parameters?

4. In your previous studies (Ref19, 20 and 21), you have developed different automated differentiation models with novel ECG parameters, such as Frontal and horizontal percent amplitude change (PAC), frontal and horizontal percent time-voltage area change (PTVAC), QRS axis change, etc. Have you tested whether the new parameters in this study outperformed or improved the diagnosis performance compare the previous models?

Here are some minor comments:

- In reference 19, 317 paired WCT (157 VT, 160 SWCT) and baseline ECGs from 213 patients. In reference 20 and 26, 601 paired WCT (273 VT, 328 SWCT) and baseline ECGs from 421 patients. Based on these data, there are more than one paired ECGs from one patient. In this training ECG dataset, it's worth to note that each patient has only one paired WCT data in this study.

- Generally, the polarity of QRS is the sum of amplitude of QR waves. Your QRS polarity is calculated as the integral of QRS waves. It's probably clearer to state that the QRS polarity is defines as by the sum (Σ) of QRS complex waveform over QRS duration in this study.

- The table (Supplemental Table S1, S2 and S3) are long and not easy to follow, maybe consider using bar graph to present

the data.

• In Result section:

WCT Polarity Code (WCT-PC): Figure 4 should be Figure 6

QRS Polarity Shift (QRS-PS): Figure 5 should be Figure 7

• It's more intuitive to combine table 3 and 4 in a bar graph, combine table S8 and S9 in a bar graph.

• In Table 1 and 2, "Patient Age" Should be in Clinical Characteristics category.

• It's more concise to move Info in Figure 4 into Figure 3.

Reviewer #2

(Remarks to the Author)

This paper proposed some ML models for Automated Differentiation of Wide QRS Complex Tachycardias. This is an interesting work. However, the models used in this work are all outdated. My suggestion is that the authors should apply some novel models in their work.

Reviewer #3

(Remarks to the Author)

The authors examined diagnostic characteristics of the QRS polarity that can be integrated into automated machine learning models aimed at achieving automated algorithms for WCTs differentiation (VT versus SWCT). The algorithms utilize (i) the direction of QRS complex polarity in WCTs, referred to as the WCT-PC (WCT Polarity Code), and (ii) the changes in QRS polarity direction between the WCT and baseline ECG, termed QRS-PS (QRS Polarity Shift). It is interesting and important topic of growing field to utilize AI in different medicine specialties and more recently in EP.

Defining equiphase (=) polarity: It is very rare to have + equal – polarity exactly in the same QRS. Any software rate 0.4-0.6 for + and 0.6-0.4 for negative? Even the examples did not show perfect + equal – polarity to call it equal.

Need some clarification: Are we comparing the tallest R to the deepest S or Q to determine is the QRS + or – or = or the summit or + r,R,r',R' to the summit of the depth of the Q and S,s,s' ?

Do we have the details of the diagnosed SWCT. I understand that comprehensive conclusion can be done for the SWCT sub-types like pre-excitation, however it will be helpful to understand which types and percentage of SWCT were identified in this study.

It seems that we are including WC baseline ECGs and not only narrow baseline ECGs? If yes, can we list the percentage for each group (VT or SWCT) and what was the WC diagnosis (LBBB, RBBB, paced, IVCD).

Can the authors provide the mean, median and SD for the VT versus SWCT cycle length and same for the QRS duration of the VT versus SWCT and change in the QRS duration from baseline to the VT versus SWCT.

Any possible variation in the leads position for baseline ECGs done the WCT (if shock was done)?

How old are the archived ECGs accepted (figure 8) from the time of the old ECGs? Any possible changes for the old ECGs and a new baseline after the WCT. Which ECG will be chosen? Any cutoff for the VT cycle length to be included in the analysis. Some very fast WCT can create some difficulties in defining the QRS polarity exactly. I am not sure if subgroup analysis can be done to see any variation in the results based on the WCT cycle length?

Figure 6: Why the SD was larger in the second institution the testing cohort when compared to the first institution (the training cohort)? Need some comments.

In figure supplement S1: 59 out of 224 excluded WCTs were due to faulty or missing baseline ECG data?

Can we have more details? If one lead was bad quality or missing the software will not be able to analyze that or the accuracy is undetermined is such patients? In the same figure: Any reason 164 WCT were excluded due to surplus ECGs. Was the excluded 164 ECGs tested to validate no difference in the VT or SWCT diagnosis with possible some slight ECG leads position variation or if the VT cycle length variation was present?

Version 1:

Reviewer comments:

Reviewer #1

(Remarks to the Author)

Thank authors for addressing and explaining for the questions and comments. Followings are a couple of more questions would like authors to look into.

1. There are a few clinical parameters: like coronary artery disease, prior myocardial infarction; ischemic cardiomyopathy, LVEF, bundle branch block, show significantly different between SWCT and VT groups. Did you consider to include the clinical parameters in your model or test your novel parameters improve the clinical model?

2. WCT QRS duration is readily available from computerized ECG measurements. Also, it is significantly different between SWCT and VT groups. Did you check whether WCT QRS duration could help the discrimination power of your model?

Reviewer #3

(Remarks to the Author)

I appreciate addressing all my concerns. I have only one new question. The BBB was more common in the SWCT than VT. Was any subgroup analysis performed to compare between patients with BBB and not BBB. One will expect that no change in polarity directions between baseline ECGs in patients with BBB and presented with WCT due to SWCT.

Version 2:

Reviewer comments:

Reviewer #1

(Remarks to the Author)

Thank you for addressing all my questions. I have no further questions.

Reviewer #3

(Remarks to the Author)

All my questions were answered and nothing to add.

SCHOOL OF MEDICINE

John T. Milliken Department of Medicine
Division of Cardiology

Dear Dr. Barnes and Dr. Cunha:

We thank you very much for the opportunity to revise our manuscript entitled “Automated Differentiation of Wide QRS Complex Tachycardias Using QRS Complex Polarity”. We also would like to recognize that the editors of Nature Communication Medicine did recruit high-quality reviewers for our manuscript. We greatly appreciate feedback, scrutiny, and discourse that this brings. Through their assistance we find that the quality of our manuscript has increased.

As requested in the decision letter, we have reviewed and responded to each point from the reviewers. We greatly thank each reviewer for their thoughtful review and commentary.

Below we have responded to each of the issues raised by the reviewers in a point-by-point manner. Specific changes to the manuscript’s text (**shown in red text**) have been detailed within the responses below.

Reviewer #1 (Remarks to the Author):

The authors of this manuscript present the results of a research that aims to develop automated differentiation of Wide QRS Complex Tachycardia to provide clinicians with estimation of ventricular tachycardia (VT) /supraventricular wide complex tachycardia (SWCT) likelihood. In this study, they described and explored innovative diagnostic characteristics that can be integrated into automated machine learning models directly derived from standard computerized ECG measurements.

They trained and tested the novel machine learning (ML) models using computerized and derived ECG data from WCT ECG alone and paired WCT and baseline ECGs on two separated datasets. The results showed that the incorporation of features derived from the WCT data alone into various ML modeling techniques resulted in favorable diagnostic performance. They also found that novel features derived from paired WCT and baseline ECG data resulted in improved diagnostic performance.

Their conclusion is that accurate VT and SWCT classification may be accomplished using readily available computerized electrocardiogram interpretation data

provided by the WCT alone or paired WCT and baseline ECGs.

Author Response:

We thank this reviewer for their particularly thorough and insightful review of our manuscript. We commend this reviewer's capacity for peer review (very high).

Reviewer Remark(s):

There are some questions would like to be discussed,

1. There are 72% and 65% of VT cases having Gold Standard Diagnosis in training and testing cohorts, and 26% and 34% of SWCT cases having Gold Standard Diagnosis. Have you thought about balancing the confirmed diagnosis rate in the two group? Such as having diagnosis agreements with experienced rhythm cardiologists.

Author Response:

We appreciate the reviewer's question and comments.

Indeed, our derivation and testing cohorts differ somewhat. This distinction arises because the cohorts were obtained from two separate institutions, each serving unique populations (Large hospital system in mid-sized city in Minnesota vs. large hospital large metropolitan area in Missouri).

We deliberately chose not to use statistical methods to homogenize the data. Instead, we considered the use of two separate insitutions as an opportunity to test the validity and generalizability of our models in the presence of 'data drift,' as the reviewer noted. Given that model performance remained strong despite this, we consider it a relative strength of our study. We consider it akin to a successful external validation.

Reviewer Remark(s):

2. How did you derive the threshold value for the polarity of QRS complex: 250

$\mu V \cdot ms$ above or below the isoelectric baseline?

Author Response:

We thank the reviewer for their question.

The threshold values were not established using a data-driven approach. Instead, they were arbitrarily chosen based on what would be unambiguously consistent with positive (+) or negative (-) if the ECG were visually inspected.

We acknowledge that using a data-driven approach to define the polarity parameter or treating Σ time-voltage area values as a continuous parameter could potentially yield better model performance. However, we think our approach helps attain explainability to the final ML model outputs and allows clinical use of the concepts outside of the diagnostic modeling. Specifically, physicians would be able to apply Bayesian principles (ie, use likelihood ratios [LRs] to determine post-test probability). For example, as a clinician reading an ECG with WCT, I can deduce that a tracing with negative QRS polarity in lead aVL increases (albeit slightly) the likelihood VT (+ LR is ~ 1.5). Alternatively, I could deduce that a ECG tracing demonstrating a polarity shift at lead V1 alone substantially increases likelihood of VT (+ LR is ~ 3).

Reviewer Remark:

3. The results in Part 3 show adding all parameters (WCT ECG data + Paired WCT and Baseline ECG data) to ML models reduced the diagnosis performance especially for LR method. Have you looked into the correlations among these parameters?

Author Response:

Thank you for this nuanced questioned.

Collinearity was an issue that was clearly apparent, especially for the logistic regression (LR) model. The presence of correlated features led to model instability, indicating parameter redundancy that did not contribute additional information and degraded model performance. This issue was especially exacerbated by increasing model complexity through the addition of more interrelated parameters in Part 3.

To confirm the presence of multicollinearity among the covariates, we calculated the Variance Inflation Factor (VIF) for the independent variables in the logistic regression model. The results showed that all selected covariates had VIF values greater than 5, indicating severe multicollinearity. This high multicollinearity likely contributed to the poorer performance of the LR model in Part 3.

In contrast, other modeling architectures, such as Random Forests (RF) and Support Vector Machines (SVMs), demonstrated a better capacity to handle interrelated parameters. Despite this, there was no significant improvement in diagnostic performance across all models when comparing Part 3 to Part 2. However, we observed that several parameters in the RF model in Part 3 were identified as non-important factors (importance score 0.00) in the model outputs, further highlighting the limitations caused by multicollinearity.

We have amended the manuscript to reflect this observation:

- Page 11, paragraph 1 and 2

“Diagnostic performance metrics of the five ML models, when applied to the training cohort, is summarized in **Supplemental Table S12. Table 5** demonstrates the diagnostic performance of the ML modeling subtypes when applied to subjects with a gold standard and non-gold standard diagnosis in the testing cohort, respectively. Comparisons of ML model performance are shown in **Supplemental Table S13**. Specific tuning parameters used for ML model training are shown in **Supplemental Table S14**. The importance of various WCT differentiation features in the RF model is shown in **Supplemental Figure S15**.

In Part 3, within the gold standard cohort, ML models displayed varying capabilities in diagnostic performance, with AUC values ranging from 0.72 to 0.93. Notably, SVM and RF methods outperformed other approaches, while the LR method was the weakest performer. Specific to the LR model, all covariates had Variance Inflation Factor values greater than 5, indicating substantial multicollinearity. In the RF model, six parameters in Part 3 were identified as non-important factors (importance score 0.00) for model outputs.”

- Page 11, paragraph 4, lines 7-10

“However, the aggregation of all available features used in Parts 1 and 2 in Part 3 did not lead to a meaningful improvement in diagnostic performance. This finding indicates a ceiling in diagnostic performance due to parameter multicollinearity.”

Reviewer Remark(s):

4. In your previous studies (Ref19, 20 and 21), you have developed different automated differentiation models with novel ECG parameters, such as Frontal and horizontal percent amplitude change (PAC), frontal and horizontal percent time-voltage area change (PTVAC), QRS axis change, etc. Have you tested whether the new parameters in this study outperformed or improved the diagnosis performance compare the previous models?

Author Response:

We thank the reviewer for their question.

In this work, we did not compare the performance of these models to previously published works. However, we have recently conducted a head-to-head performance comparison between five automated WCT differentiation models (WCT Formula, WCT Formula II, VT Prediction Model, Solo Model, and Paired Model) and three manual interpretation methods (Brugada algorithm, Vereckei aVR algorithm, VT score). This work has already been accepted and published in *Circulation: Arrhythmia and Electrophysiology* (1).

The main intent of this work is to introduce novel predictive features that may be incorporated into new modeling approaches for WCT differentiation. While we do plan to formally report a comparison between these newer models and those of prior works in the future, we have not included such an analysis in this study due to its existing length and already high data amount. Furthermore, our primary interest lies in eventually formulating a ‘comprehensive model’ that integrates the engineered features from this work with other highly predictive features already described.

We have amended the manuscript.

- Page 14, paragraph 3, lines 15-19

“Recently, our group published a separate analysis comparing the performance of novel automated models with traditional manual WCT differentiation approaches (1). We observed automated models demonstrated favorable performance compared to manual WCT differentiation methods across multiple diagnostic metrics.

Reviewer Remark(s):

Here are some minor comments:

In reference 19, 317 paired WCT (157 VT, 160 SWCT) and baseline ECGs from 213 patients. In reference 20 and 26, 601 paired WCT (273 VT, 328 SWCT) and baseline ECGs from 421 patients. Based on these data, there are more than one paired ECGs from one patient. In this training ECG dataset, it's worth to note that each patient has only one paired WCT data in this study.

Author Response:

Thank you for this point. We agree and have amended the manuscript accordingly.

- Page 5, paragraph 3, lines 7-9

“Among patients with multiple WCT events, any subsequent WCT ECGs occurring after the first event were excluded, ensuring that each selected patient had only one pair of WCT and baseline ECGs for analysis.”

Reviewer Remark(s):

Generally, the polarity of QRS is the sum of amplitude of QR waves. Your QRS polarity is calculated as the integral of QRS waves. It's probably clearer to state that the QRS polarity is defines as by the sum (Σ) of QRS complex waveform over QRS duration in this study.

Author Response:

Thank you for this comment. We appreciate this feedback as it enables modifications for enhanced clarity. We have revised the manuscript.

- Page 7, paragraph 3, lines 3-5

“In other words, QRS polarity is determined by the cumulative summation of all QRS waveform integrals, which corresponds to the area under the QRS waveforms.”

Reviewer Remark(s):

The table (Supplemental Table S1, S2 and S3) are long and not easy to follow, maybe consider using bar graph to present the data.

Author Response:

We appreciate the reviewer's feedback. We acknowledge that the data tables in Supplemental Tables S1, S2, and S3 are extensive and may be challenging to follow. While we understand the suggestion to present the data with bar graphs instead, we opted for tables as we felt they provided the most comprehensive and easily interpretable way to present the data. This is because transitioning to bar graphs would result in a very large number of bars (72 bars for Table S1 and S2, and 216 bars for Table S3), which we believe could potentially complicate rather than simplify the data interpretation. We hope the reviewer understands our decision in this regard.

Reviewer Remark(s):

- *In Result section:
WCT Polarity Code (WCT-PC): Figure 4 should be Figure 6
QRS Polarity Shift (QRS-PS): Figure 5 should be Figure 7*

Author Response:

We thank the reviewer for their careful review. We have corrected the mistake.

Reviewer Remark(s):

- *It's more intuitive to combine table 3 and 4 in a bar graph, combine table S8 and S9 in a bar graph.*

Author Response:

We thank the reviewer for their suggestion.

In response, we have made the following changes that we believe will be satisfactory to the referee. We have added three new bar graphs to be presented alongside Tables 3, 4, and 5. We did not similarly do the same for the data in Supplemental Tables due to their lesser degree of importance. If there are space constraints or concerns from the editorial staff, we believe that the bar graphs alone we acceptably convey the manuscripts data.

Reviewer Remark(s):

- *In Table 1 and 2, “Patient Age” Should be in Clinical Characteristics category.*

Author Response:

We thank the reviewer for their careful review. We have corrected the mistake.

Reviewer Remark(s):

- *It’s more concise to move Info in Figure 4 into Figure 3.*

Author Response:

We thank the reviewer for their excellent suggestion.

We have merged the content of the figures into one figure. Please see our new Figure 3 in the revised manuscript.

Reviewer #2 (Remarks to the Author):

This paper proposed some ML models for Automated Differentiation of Wide QRS Complex Tachycardias. This is an interesting work. However, the models used in this work are all outdated. My suggestion is that the authors should apply some novel models in their work.

Author Response:

We thank the reviewer for their review and comment. We also appreciate their courteous feedback.

We presume the reviewer's point is regarding the use of deep learning models for the task of WCT differentiation. While we agree that this form of machine learning is more advanced, we argue that it is not necessarily more effective for our specific task. In fact, based on our previous attempts with deep learning architectures (unpublished research) for WCT differentiation, we found that their performance to be underwhelming, leading us to pursue alternative ML approaches

Reviewer #3 (Remarks to the Author):

The authors examined diagnostic characteristics of the QRS polarity that can be integrated into automated machine learning models aimed at achieving automated algorithms for WCTs differentiation (VT versus SWCT). The algorithms utilize (i) the direction of QRS complex polarity in WCTs, referred to as the WCT-PC (WCT Polarity Code), and (ii) the changes in QRS polarity direction between the WCT and baseline ECG, termed QRS-PS (QRS Polarity Shift). It is interesting and important topic of growing field to utilize AI in different medicine specialties and more recently in EP.

Author Response:

We thank the reviewer for the thoughtful review and commentary. We easily recognize and commend this reviewer's advanced clinical expertise.

We agree that this area of research will be pivotal in shaping the future of diagnostic electrocardiography, a foundational medical science for an ubiquitous diagnostic modality integral to nearly all healthcare disciplines. We also believe that offering new solutions for accurate WCT differentiation may significantly improve patient care, especially in high-acuity settings such as intensive care units and emergency rooms.

Reviewer Remark(s):

Defining equiphasic (=) polarity: It is very rare to have + equal – polarity exactly in the same QRS. Any software rate 0.4-0.6 for + and 0.6-0.4 for negative? Even the examples did not show perfect + equal – polarity to call it equal.

Author Response:

We agree with reviewers point, and we appreciate the opportunity to clarify this point.

In this work, we defined equiphasic QRS complexes as those where the sum of the waveforms falls within the range of $250 \mu\text{V}\cdot\text{ms}$ above and below the isoelectric baseline (i.e., $-250 \mu\text{V}\cdot\text{ms} \leq \Sigma \text{QRS TVA} \leq 250 \mu\text{V}\cdot\text{ms}$).

We have amended the manuscript to make this point more clear.

- Page 7, paragraph 3, lines, 13-16

“If the sum of the QRS complex waveforms falls within the range of $250 \mu\text{V}\cdot\text{ms}$ above or below the isoelectric baseline (i.e., $-250 \mu\text{V}\cdot\text{ms} \leq \Sigma \text{QRS TVA} \leq 250 \mu\text{V}\cdot\text{ms}$), the QRS complex is defined as having equiphasic (=) polarity.”

Reviewer Remark(s):

Need some clarification: Are we comparing the tallest R to the deepest S or Q to determine is the QRS + or – or = or the summit or + r,R,r',R' to the summit of the depth of the Q and S,s,s' ?

Author Response:

We thank the reviewer for their question.

For this analysis, positive (+), negative (-), and equiphasic (=) QRS polarity were determined by the sum (Σ) of QRS complex waveform TVAs ($\mu\text{V}\cdot\text{ms}$) above (r/R and r'/R') and below (q/QS, s/S, and s'/S') the isoelectric baseline. In other words, QRS polarity is determined by the cumulative summation of all QRS waveform integrals, which corresponds to the area under the QRS waveforms. We have amended the manuscript to make this point more clear.

- Page 7, paragraph 2 lines, 3-6

“In other words, QRS polarity is determined by the cumulative summation of all QRS waveform integrals, which corresponds to the area under the QRS waveforms.”

Reviewer Remark(s):

Do we have the details of the diagnosed SWCT. I understand that comprehensive conclusion can be done for the SWCT sub-types like pre-excitation, however it will be helpful to understand which types and percentage of SWCT were identified in this study.

Author Response:

We thank the reviewer for their question.

Unfortunately, we do not have this information. Regrettably, the archiving of more granular details regarding the electrophysiologic nature of various rhythms (such as the mapping of accessory pathways or the site of origin for VT) is not readily available in our electronic medical records. Moreover, it is challenging to confidently connect intricate details of EP procedure results with observations on the 12-lead ECG obtained separately in real-world clinical scenarios, especially if it is more than a general description of the underlying rhythm (VT vs. SWCT). Therefore, beyond establishing general rhythm classifications, we are unable to conduct a more precise analysis.

Reviewer Remark(s):

It seems that we are including WC baseline ECGs and not only narrow baseline ECGs? If yes, can we list the percentage for each group (VT or SWCT) and what was the WC diagnosis (LBBB, RBBB, paced, IVCD).

Author Response:

We thank the reviewer for their request.

We have included additional information in revised Tables 1 and 2 of the revised manuscript.

Reviewer Remark(s):

Can the authors provide the mean, median and SD for the VT versus SWCT cycle length and same for the QRS duration of the VT versus SWCT and change in the QRS duration from baseline to the VT versus SWCT.

Author Response:

We thank the reviewer for their question.

We cannot provide numbers re: cycle length, as they are not standard measurements provided by the computerized ECG interpretation software.

As it relates to the question of changes in QRS duration, we have published on this in the past. We included a table from a prior work (2) we trust the reviewer find to be interesting (see below). The pearl learned from this data is that large QRS duration changes (irrespective of the baseline QRS duration) is highly predictive of VT. So much so, we have used such a paramter in some of our previously published models.

Table 2
Electrocardiographic variables among baseline ECG sub-groups.^a

Electrocardiographic measurements	Baseline QRS duration <120 ms VT (n = 54) SWCT (n = 42)			Baseline QRS duration ≥ 120 ms VT (n = 103) SWCT (n = 118)			Baseline ventricular pacing VT (n = 69) SWCT (n = 10)		
	VT	SWCT	p value	VT	SWCT	p value	VT	SWCT	p value
WCT QRS duration (ms)	171.1 (33.1)	143.1 (19.9)	<0.001	180.2 (31.8)	144.9 (17.5)	<0.001	187.2 (26.4)	157.2 (17.7)	<0.001
QRS duration change (ms)	71.5 (32.4)	41.5 (21.1)	<0.001	31.3 (26.3)	11.2 (16.8)	<0.001	37.5 (30.7)	23.0 (45.0)	0.19
QRS axis change (°)	93.7 (50.5)	45.1 (47.6)	<0.001	76.2 (59.1)	17.8 (21.2)	<0.001	90.2 (58.2)	26.6 (26.7)	<0.001
Frontal PAC (%)	116.2 (59.0)	47.0 (25.1)	<0.001	128.4 (98.7)	30.6 (29.4)	<0.001	135.8 (94.6)	61.9 (75.6)	0.004
Horizontal PAC (%)	128.8 (72.6)	57.9 (26.7)	<0.001	108.2 (56.3)	40.1 (23.0)	<0.001	123.6 (66.4)	49.2 (25.4)	<0.001

^a Standard deviation is in parentheses. PAC = percent amplitude change; SWCT = supraventricular wide complex tachycardia; VT = ventricular tachycardia; WCT = wide complex tachycardia.

Reviewer Remark(s):

Any possible variation in the leads position for baseline ECGs done the WCT (if shock was done)? How old are the archived ECGs accepted (figure 8) from the time of the old ECGs? Any possible changes for the old ECGs and a new baseline after the WCT. Which ECG will be chosen? Any cutoff for the VT cycle length to be included in the analysis. Some very fast WCT can create some difficulties in defining the QRS polarity exactly. I am not sure if subgroup analysis can be done to see any variation in the results based on the WCT cycle length?

Author Response:

We appreciate the reviewer's insightful questions regarding ECG selection. Our responses to each are detailed below.

Baseline ECGs were defined as the first non-WCT rhythm recorded after the WCT event in the training cohort. In the testing cohort, the most proximate non-WCT rhythm to the WCT event was defined as the baseline ECG (acquired before or after the WCT event). We have amended the manuscript to clarify this point.

- Page 5, paragraph 2 lines, 7-9

“Baseline ECGs were defined as the (i) first non-WCT rhythm recorded *after* the WCT event (Training cohort [Institution #1]) or (ii) most proximate non-WCT rhythm to the WCT event (Testing cohort [Institution #2]).”

Due to their acquisition at different time points in genuine clinical practice, minor lead placement variations were inevitable for most ECGs analyzed. We have amended the manuscript to add data re: the time separation between paired baseline and WCT ECGs.

- Page 9, paragraph 2, lines 10-11

“The median time between the WCT and baseline ECG was 5.6 hours (interquartile range: 1.0, 41.8).”

- Page 9, paragraph 3, lines, 7-8

“The median time between the WCT and baseline ECG was 16.2 hours (interquartile range: 1.9, 67.5).”

Lastly, we agree that QRS polarity can be difficult to determine for very fast rhythms. However we did not have have any cut-off determined by cycle-length or its corollary heart rate. As such, we also do not have the data to perform a subgroup analysis base on WCT cycle length.

Reviewer Remark(s):

Figure 6: Why the SD was larger in the second institution the testing cohort when compared to the first institution (the training cohort)? Need some comments.

Author Response:

We thank the reviewer for their question.

The differences in SD between cohorts were because there were less ECGs in second institution (Testing cohort) than the first institution (Training cohort).

Reviewer Remark(s):

In figure supplement S1: 59 out of 224 excluded WCTs were due to faulty or missing baseline ECG data? Can we have more details? If one lead was bad quality or missing the software will not be able to analyze that or the accuracy is undetermined is such patients? In the same figure: Any reason 164 WCT were excluded due to surplus ECGs. Was the excluded 164 ECGs tested to validate no difference in the VT or SWCT diagnosis with possible some slight ECG leads position variation or if the VT cycle length variation was present?

Author Response:

We thank the reviewer for their excellent questions.

Regarding the reviewer's first query, the excluded ECG pairs were primarily due to high-frequency artifacts, including multiple CRT pacing spikes in the baseline ECG and LVAD motor interference affecting both WCT and baseline ECGs. These issues occasionally resulted in incomplete or inaccurate QRS complex measurement data for both baseline and WCT analyses. In such instances, proprietary computerized ECG programs did not perform as expected, highlighting internal flaws in the manufacturer's product under specific conditions. We have discussed this very issue with the Chief scientist and other engineers at *GE Healthcare*.

Regarding the reviewer's second point, in previous studies, we treated multiple ECG pairs from a single patient as independent events due to their common occurrence in clinical practice. However, for this study, we chose not to do so to avoid potential bias in modeling performance from patients with multiple WCT events that essentially represented the same underlying rhythm. This approach was the preferred approach for our data scientist co-authors. That being stated, I find that both approaches have their merits, but for the purposes of this investigation we opted to exclude surplus ECG pairs.

We sincerely appreciate comments and feedback provided by the reviewers. We believe the manuscript has been strengthened with the aforementioned changes. A revised submission has been submitted for your continued consideration.

This manuscript is not under consideration for publication elsewhere and has not been previously published.

All authors have made significant contributions to the study and are thoroughly familiar with all data; all authors are responsible for the contents enclosed, including the manuscript and subsequent revision.

Sincerely,

Adam May, MD

References

1. LoCoco S, Kashou AH, Deshmukh AJ et al. Direct Comparison of Methods to Differentiate Wide Complex Tachycardias: Novel Automated Algorithms Versus Manual ECG Interpretation Approaches. *Circ Arrhythm Electrophysiol* 2024:e012663.
2. May AM, DeSimone CV, Kashou AH et al. The WCT Formula: A novel algorithm designed to automatically differentiate wide-complex tachycardias. *J Electrocardiol* 2019;54:61-68.

SCHOOL OF MEDICINE

John T. Milliken Department of Medicine
Division of Cardiology

Dear Reviewers,

Thank you for the opportunity to address your comments on our manuscript titled "Automated Differentiation of Wide QRS Complex Tachycardias Using QRS Complex Polarity."

We have carefully reviewed and responded to each point raised, and we greatly appreciate your thoughtful feedback and critiques throughout this review process.

In this rebuttal, we address each issue in detail. Please note that we did not make specific changes to the manuscript text, as there did not appear to be a direct request for modifications.

Reviewer #1 (Remarks to the Author):

Reviewer Remark(s):

Thank authors for addressing and explaining for the questions and comments. Followings are a couple of more questions would like authors to look into.

Author Response:

We thank this reviewer for their continued review of our manuscript. Again, we thank this reviewer for their courteous feedback and insightful comments and questions.

Reviewer Remark(s):

There are a few clinical parameters: like coronary artery disease, prior myocardial infarction; ischemic cardiomyopathy, LVEF, bundle branch block, show significantly different between SWCT and VT groups. Did you consider to include the clinical parameters in your model or test your novel parameters improve the clinical model?

Author Response:

We appreciate the reviewer's fantastic question.

Our group has considered incorporated such information into automated or semi-automated WCT differentiation models. We designed a separate project for this purpose, and we will develop a newer and more robust model that incorporates clinical information. We envision that it will function similarly to the Mayo Clinic VT Calculator on a webbased platform, or as a tool built into the electronic health record.

Reviewer #3 (Remarks to the Author):

Reviewer Remark(s):

I appreciate addressing all my concerns.

Author Response:

You are most welcome. And we wish to personally thank the reviewer for the continued evaluation of our manuscript.

Reviewer Remark(s):

I have only one new question. The BBB was more common in the SWCT than VT. Was any subgroup analysis performed to compare between patients with BBB and not BBB. One will expect that no change in polarity directions between baseline ECGs in patients with BBB and presented with WCT due to SWCT.

Author Response:

Thank you for the excellent and nuanced question. We agree with the hypothesis that polarity changes in patients with a baseline BBB strongly suggest VT and are likely more pronounced than in patients without a baseline BBB.

Although we did not conduct a formal analysis for this manuscript due to the complexity of evaluating multiple parameters (12 for each ECG lead) and covariates (three for polarity shift, partial shift, and no shift) in groups with and without baseline BBB, we are pleased to provide an informal alternative for the sake of addressing the reviewer's question and offering evidence to support their hypothesis (see below). For simplicity, this analysis focuses on frontal and horizontal percent

amplitude change (PAC), which, like QRS polarity shifts, reflect electrophysiological differences between VT and SWCT. Higher PAC values are associated with VT, while lower values indicate SWCT.

An analysis of 601 WCT and baseline ECG pairs from Institution #1 (Mayo Clinic) demonstrated significant differences in frontal and horizontal PAC between VT and SWCT groups, regardless of baseline BBB presence. The logistic regression model's performance with these parameters indicated improved diagnostic accuracy in pairs with a baseline BBB. However, direct comparisons of model performance were not feasible due to differences in subgroup composition.

WCT and Baseline ECG pairs with a baseline BBB			
	VT (n = 39)	SWCT (n = 217)	
Median Frontal PAC (%)	107.9%	25.0%	p-value < 0.001
Median Horizontal PAC (%)	87.8%	33.5%	p-value < 0.001
2-parameter logistic regression model performance (203 WCT and baseline ECG pairs in derivation cohort, 58 WCT and baseline pairs in validation cohort)			AUC 0.923

WCT and Baseline ECG pairs without a baseline BBB			
	VT (n = 233)	SWCT (n = 111)	
Median Frontal PAC (%)	109.6%	40.6%	p-value < 0.001
Median Horizontal PAC (%)	110.3%	47.5%	p-value < 0.001
2-parameter logistic regression model (253 WCT and baseline ECG pairs in derivation cohort, 91 WCT and baseline pairs in validation cohort)			AUC 0.882

We sincerely appreciate the comments, questions, and feedback provided throughout this process.

This manuscript is not under consideration for publication elsewhere and has not been previously published.

All authors have made significant contributions to the study, are thoroughly familiar with the data, and take full responsibility for the content of the enclosed materials, including the revised manuscript.

Sincerely,

Adam May, MD

SCHOOL OF MEDICINE

John T. Milliken Department of Medicine
Division of Cardiology

Dear Dr. Barnes and Dr. Cunha,

Thank you for the opportunity to review the reviewers' comments on our manuscript titled "Automated Differentiation of Wide QRS Complex Tachycardias Using QRS Complex Polarity."

We have reviewed the final input and comments by our reviewers, and it appears they were satisfied by our most recent manuscript revision and rebuttal.

Reviewer #1 (Remarks to the Author):

Reviewer Remark(s):

All my questions were answered and nothing to add.

Author Response:

We greatly thank this reviewer for their review our manuscript.

Reviewer #3 (Remarks to the Author):

All my questions were answered and nothing to add.

Author Response:

We greatly thank this reviewer for their review our manuscript.

Again, we sincerely appreciate comments, questions, and feedback provided by the reviewers. We also want to thank the editorial staff for their expert handling of our manuscript.

This manuscript is not under consideration for publication elsewhere and has not been previously published.

All authors have made significant contributions to the study and are thoroughly familiar with all data; all authors are responsible for the contents enclosed, including the revised manuscript.

Sincerely,

Adam May, MD